# Low Zinc Levels at Admission Associates with Poor Clinical Outcomes in SARS-CoV-2 Infection

**DOI:** 10.3390/nu13020562

**Published:** 2021-02-09

**Authors:** Marina Vogel-González, Marc Talló-Parra, Víctor Herrera-Fernández, Gemma Pérez-Vilaró, Miguel Chillón, Xavier Nogués, Silvia Gómez-Zorrilla, Inmaculada López-Montesinos, Isabel Arnau-Barrés, Maria Luisa Sorli-Redó, Juan Pablo Horcajada, Natalia García-Giralt, Julio Pascual, Juana Díez, Rubén Vicente, Robert Güerri-Fernández

**Affiliations:** 1Laboratory of Molecular Physiology, Department of Experimental and Health Sciences, Universitat Pompeu Fabra, 08003 Barcelona, Spain; marina.vogel@upf.edu (M.V.-G.); victor.herrera01@estudiant.upf.edu (V.H.-F.); ruben.vicente@upf.edu (R.V.); 2Molecular Virology Group, Department of Experimental and Health Sciences, Universitat Pompeu Fabra, 08003 Barcelona, Spain; marc.tallo@upf.edu (M.T.-P.); gemma.perez@upf.edu (G.P.-V.); juana.diez@upf.edu (J.D.); 3Department of Biochemistry and Molecular Biology and Institute of Neurosciences, Edifici H, Universitat Autònoma de Barcelona, 08193 Bellaterra, Spain; Miguel.Chillon@uab.cat; 4Unitat Mixta UAB-VHIR, Vall d’Hebron Institut de Recerca (VHIR), 08035 Barcelona, Spain; 5Institut Català de Recerca i Estudis Avançats (ICREA), 08010 Barcelona, Spain; 6Department of Internal Medicine, Hospital del Mar, Institut Mar d’Investigacions Mediques, 08003 Barcelona, Spain; xnogues@psmar.cat (X.N.); ngarcia@imim.es (N.G.-G.); 7Department of Infectious Diseases, Hospital del Mar, Institut Mar d’Investigacions Mediques, 08003 Barcelona, Spain; SGomezZorrilla@parcdesalutmar.cat (S.G.-Z.); ilopezmontesinos@psmar.cat (I.L.-M.); iarnau@psmar.cat (I.A.-B.); lsorli@psmar.cat (M.L.S.-R.); jhorcajada@psmar.cat (J.P.H.); 8Department of Nephrology, Hospital del Mar, Institut Mar d’Investigacions Mediques, Autonomous University of Barcelona, 08003 Barcelona, Spain; JPascualSantos@parcdesalutmar.cat

**Keywords:** SARS-CoV-2, zinc, clinical outcomes

## Abstract

Background: Zinc is an essential micronutrient that impacts host–pathogen interplay at infection. Zinc balances immune responses, and also has a proven direct antiviral action against some viruses. Importantly, zinc deficiency (ZD) is a common condition in elderly and individuals with chronic diseases, two groups with an increased risk for severe severe coronavirus disease 2019 (COVID-19) outcomes. We hypothesize that serum zinc content (SZC) influences COVID-19 disease progression, and thus might represent a useful biomarker. Methods: We ran an observational cohort study with 249 COVID-19 patients admitted in Hospital del Mar. We have studied COVID-19 severity and progression attending to SZC at admission. In parallel, we have studied severe acute respiratory syndrome coronavirus 2 (SARS-CoV2) replication in the Vero E6 cell line modifying zinc concentrations. Findings: Our study demonstrates a correlation between serum zinc levels and COVID-19 outcome. Serum zinc levels lower than 50 µg/dL at admission correlated with worse clinical presentation, longer time to reach stability, and higher mortality. Our in vitro results indicate that low zinc levels favor viral expansion in SARS-CoV-2 infected cells. Interpretation: Low SZC is a risk factor that determines COVID-19 outcome. We encourage performing randomized clinical trials to study zinc supplementation as potential prophylaxis and treatment with people at risk of zinc deficiency.

## 1. Introduction

Infections with severe acute respiratory syndrome coronavirus 2 (SARS-CoV-2) result in a systemic disease with a variety of outcomes, from no symptoms to severe and diverse pathologies, with pneumonia and acute respiratory distress symptom being the most common. Therefore, a better knowledge of risk factors determining severity is essential for treating patients in the early stages of COVID-19.

Zinc (Zn^2+^) is an essential trace element required for maintaining a variety of fundamental biological processes, due to its functions as a cofactor, signaling molecule, and structural element. One of the most significant roles of zinc in our body is its broad effect on the immune system [1,2], as both the adaptive and the innate immunity, are affected by zinc levels. Consequently, zinc deficiency (ZD), due to low zinc intake or malabsorption, results in an immune imbalance that ultimately causes a major public health problem with high prevalence in elderly and individuals with chronic diseases [1]. In adaptive immunity, zinc affects T lymphocyte maturation, differentiation, and cytokine production. B cell activation and plasma cell differentiation depend on zinc signaling as well [2,3]. In innate immunity, zinc has an anti-inflammatory role [4]. Concretely, ZD is associated with higher levels of interleukin (IL)-1beta and tumor necrosis factor (TNF)-alfa [5], as well as with altered activities of monocytes, neutrophils, and natural killer (NK) cells [4,6,7]. Correspondingly, ZD results in increased susceptibility to inflammatory and infectious diseases, including acquired immune deficiency syndrome, measles, malaria, tuberculosis, and pneumonia [2].

In this context, zinc supplementation trials have been carried out to reduce morbidity and mortality in developing countries with high ZD incidence. Importantly, zinc supplementation significantly reduced the duration of respiratory tract infections caused by rhinoviruses and coronaviruses [8]. Moreover, zinc has been shown to have a direct antiviral action [9]. Remarkably, in vitro zinc inhibited the binding of the SARS coronavirus RNA-dependent RNA polymerase to its template and the subsequent elongation, as well as viral replication in a cell culture [10]. Altogether, zinc status might condition COVID-19 severity, and zinc supplementation could be a useful tool to impact COVID-19 outcome.

In this work, we focus our observational study on the putative association between the zinc status of hospitalized COVID-19 patients with disease progression and clinical outcomes. Moreover, we address in cell culture the potential of zinc supplementation to directly block SARS-CoV-2 multiplication in Vero E6 infected cells.

## 2. Materials and Methods

### 2.1. Study Design and Participants

An observational cohort study was performed at Hospital del Mar in Barcelona (Spain). This is a 400-bed, tertiary university hospital in Barcelona that provides healthcare to an urban area of 500,000 people. All patients admitted with COVID-19 for ≥48 h between 9 March and 1 April 2020 were included. COVID-19 was defined as a SARS-CoV-2 infection, confirmed by quantitative PCR (qPCR) performed in nasopharyngeal samples obtained by trained personnel at hospital admission, and by fulfilling clinical diagnostic criteria. These include any of the following: respiratory symptoms (dyspnea, cough, sore throat, changes in taste/smell) and chest X-ray findings (uni-or bilateral interstitial infiltrates) that made the diagnosis probable in the current epidemiological situation. The Institutional Ethics Committee of Hospital del Mar of Barcelona approved the study, and due to the nature of the retrospective data review, waived the need for informed consent from individual patients (CEIm-2020/9352).

### 2.2. Procedures

Demographic, clinical, epidemiological, and the whole-episode (laboratory workup, vital signs, treatment) data were extracted from electronic medical records using a standardized data collection method. Comorbidity was collected individually and as the Charlson Comorbidity Index, a clinical score that relates long term-mortality to patient comorbidity. It is considered the absence of comorbidity to be 0–1 points and high at >3 points.

Laboratory workups were systematized with an at-admission protocol that included a fasting blood draw, complete kidney and liver profile, electrolytes, blood count, coagulation profile, inflammatory markers (interleukin-6 (IL-6), serum ferritin), D-dimer, and myocardial enzymes. Zinc and selenium levels were detected by electrochemiluminescence. Follow-up laboratory workups in the clinical setting included C-reactive protein and IL-6, which made possible to determine the IL-6 peak.

### 2.3. Definitions

#### 2.3.1. Time to Recovery (TR)

Time to recovery (TR) or time to clinical stability was defined as the time elapsed since the patient’s admission to oxygen saturation >94% (FiO_2_ = 21%), normalized level of consciousness (baseline), heart rate <100 rpm, systolic blood pressure >90 mm Hg, and body temperature <37.2 °C.

#### 2.3.2. Clinical Severity

Clinical severity was assessed at admission with a modified early warning score (MEWS) [11]. The same score was used for the follow-up during the admission.

#### 2.3.3. Low Zinc Levels

Since levels of zinc have a diurnal variation, all samples were collected through a fasting blood draw at 8 am (±2 h). In our cohort, the Q1 was 50 µg/dL (7.6 µM). According to previous publications, 50 µg/dL is a proper threshold of abnormally lower zinc concentration to predict clinical signs [12]. We considered this as a predictive factor the lower range (<50 µg/dL).

### 2.4. Cell Culture

Vero E6 cells were grown as described [13,14]. When indicated, fetal bovine serum (FBS) was incubated according to the manufacturer’s instructions with Chelex 100 resin (Bio-Rad Laboratories, Hercules, CA, USA) to generate Zn^2+^-free growth medium. ZnSO_4_ was added as needed to the final medium to generate specific Zn^2+^ concentration conditions. Chloroquine (Supelco, Darmstadt, Germany) was prepared in water at 10 mM and used at the desired concentration.

### 2.5. Zinc Measurements

Cells were seeded and grown in multi-well 24 plates until reaching 80% of confluence. Cells were incubated with 1 µM of FluoZin-3AM (Invitrogen, Darmstadt, Germany) or 25 µM of Zinquin (Sigma-Aldrich, Darmstadt, Germany) for 30 min at 37 °C (5% CO_2_) in isotonic solution containing (in mM) 140 NaCl, 5 KCl, 1.2 CaCl_2_, 0.5 MgCl_2_, 5 glucose, and 10 Hepes (300 milliosmoles/liter, pH 7.4), plus different concentrations of Zn^2+^ and/or chloroquine (CQ).Cells were then dissociated with Trypsin 0.05% in 0.53 mM EDTA, and were washed with PBS 1x. Fluorescence was quantified using an LSRII flow cytometer. Further analysis was performed using Flowing software (Perttu Terho, Turun yliopisto, Turku, Finland).

For in vivo confocal imaging, cells grown on 22 mm coverslips were incubated with Lysotracker, together with FluoZin-3AM or Zinquin, in a solution with 50 µM Zn^2+^ for 30 min; they were then washed twice with PBS and placed under the microscope in isotonic solution for imaging with an SP8 Leica microscope (Wetzlar, Germany).

### 2.6. Viability Assays

Cells were exposed to different Zn^2+^ and CQ concentrations for 48 h. Then, 3-(4,5-dimethylthiazol-2-yl)-2,5-diphenyltetrazolium bromide (MTT) reagent was added to obtain a final concentration of 0.5 mg/mL. Cells were incubated 2–3 h at 37 °C. After that, the supernatant was removed, and cells were resuspended in 100 µL of DMSO. The absorbance was read at 570 nm.

### 2.7. Virus Infection and Quantification

Vero E6 cells were infected with SARS-CoV-2 strain hCoV-19/Spain/VH000001133/2020 (EPI_ISL_418860), and 48 h later viral RNA from the supernatant was extracted using the Quick-RNA Viral Kit (Zymo Research, Irvine, CA, USA). SARS-CoV-2 production was quantified by qPCR with the qScript XLT One-Step RT-qPCR ToughMix, ROX (Quanta Biosciences, Berverly, USA), using the specific probe 2019-nCoV_N1-P, 5′-FAM-ACCCCGCATTACGTTTGGTGGACC-BHQ1-3′; as well as primers 2019-nCoV_N1-F, 5′-GACCCCAAAATCAGCGAAAT-3′; and 2019-nCoV_N1-R, 5′-TCTGGTTACTGCCAGTTGAATCTG-3′ (Biomers, Ulm, Germany).

### 2.8. Western Blot

Cells were treated with 0, 10, or 50 μM of Zn^2+^ and 10 μM of CQ for 24 h. Then, cells were washed twice with cold PBS and lysed with 35 μL of lysis buffer (50 mM Tris-HCl, pH 7.4, 150 mM NaCl, 5 mM EDTA, 0.5% NP-40, 1 mM DTT, 10 mM 13-GP, 0.1 mM Na_3_VO_4_, and protease inhibitors). Lysates were vortexed for 30 min at 4 °C and centrifuged at 10,000× *g* to remove aggregates. Lysates were boiled at 95 °C and placed in ice for 1 min. 20 μL of each sample were loaded onto a 12% or 14% polyacrylamide gel. After transfer, membranes were blocked with 5% milk in TBS-Tween 0.1% for 1 h at room temperature. Primary antibodies were diluted in blocking solution—microtubule-associated proteins 1A/1B light chain 3B (LC3) (L8918, Sigma-Aldrich, Darmstadt, Germany) at 1:500, p62 (ab155686, Abcam, Cambridge, UK) at 1:1000, and glyceraldehyde 3-phosphate dehydrogenase (GAPDH) (ab8245, Abcam, Cambridge, UK) at 1:1000—and incubated overnight at 4 °C. Anti-rabbit or anti-mouse horseradish peroxidase (HRP) secondary antibodies (1:1000; GE Healthcare, Chicago, IL, USA) were used.

### 2.9. Statistical Analysis

A descriptive analysis of the main demographic characteristics of the cohort and the main clinical (laboratory, treatment and outcome) characteristics of the episode was done. Continuous and categorical variables were presented as median (interquartile range (IQR)) and absolute number (percentage), respectively. A Mann–Whitney U-test, χ^2^ test, and Fisher’s exact test were used to compare differences between individuals with serum zinc levels above and below 50 µg/mL. Bi-variate comparisons and a multiple logistic regression model studying the impact of at-admission serum zinc in in-hospital mortality were fitted adjusting by age, sex, Charslon comorbidity index, and severity of the episode. Taking into consideration the number of deaths observed, and to avoid overfitting the model, we selected the most significant variables and those that provided a more general characterization of the individuals. We excluded variables if they had collinearity with MEWS or age. We created exploratory alternative models, including previously well-known individual mortality risk factors, such as chronic kidney disease, lung disease, or diabetes. We selected the final model adjusted by age, sex, severity, and Charslon comorbidity index. Significance was a *p*-value <0.05. Stata 14 software was used.

In the in vitro assays, we applied either an unpaired Student’s *t*-test (when comparing two conditions), or one-way analysis of variance (ANOVA) followed by a Bonferroni post-hoc test (when comparing control with any other condition) was applied. All plots comparisons and *p* values are described in figure legends. Analysis was performed using GraphPad software (San Diego, CA, USA).

## 3. Results

### 3.1. Influence of SZC in COVID-19 Outcome

We included 249 consecutive adults admitted to the COVID-19 unit between 9 March and 1 April 2020. The median age of participants was 65 (54–75) years, and 49% were female. Table 1 shows the baseline characteristics of the cohort and the differences between individuals with SZC <50 µg/dL and ≥50 µg/dL at onset. Hypertension was the most prevalent comorbidity. At admission, the overall severity according to MEWS score was 2 (1–3). Fever, cough, and dyspnea were the most common symptoms, with frequencies of 203 (81%), 198 (79%), and 151 (60%), respectively. Inflammation was a hallmark of the episode, with 224 (87.5%) individuals having abnormal IL-6 at admission and with a median IL-6 of 42 pg/mL (15–89), a median C-reactive protein of 7.5 mg/dL (3.5–15.2), and a median D-Dimer of 800 UI/l (470–1450). Almost all the individuals admitted during this period received hydroxychloroquine. Other treatments were tocilizumab, dexamethasone, and methylprednisolone, each of which were prescribed in a quarter of individuals. Seventy patients (28%) were admitted to the intensive care unit (ICU), and 21 (9%) patients died during hospitalization in this period.

SZC median at admission was 61 µg/dL (9.3 µM), with 58 (23%) of the individuals presenting SZC < 50 µg/dL. Individuals with serum zinc levels <50 µg/dL had higher prevalence of chronic kidney disease and chronic respiratory disease (9% vs. 2%, *p* = 0.024, and 16% vs. 7%, *p* = 0.041, respectively). Moreover, this group had a more severe clinical presentation MEWS score (2 (2–3) vs. 2 (1–3); *p* = 0.005) and significantly greater inflammation, measured by the nonspecific marker C-reactive protein (14.6 mg/dL vs. 7 mg/dL; *p* = 0.03) and the more specific IL-6 (77 pg/mL vs. 32 pg/mL; *p* < 0.001). At admission, SZC correlated negatively with inflammation measured by IL-6 (Pearson’s *r* = −0.307; *p* < 0.001) (Figure 1A), and interestingly, also correlated negatively with the highest value of IL-6 during the episode (Pearson’s *r* = −0.317; *p* < 0.001) (Figure 1B). This negative correlation was also observed with the nonspecific inflammatory markers, such as ferritin (Pearson’s *r* = −0.227; *p* = 0.001) and C-reactive protein (Pearson’s *r* = −0.315; *p* < 0.001) (Figure 1C). We also found a correlation between SZC at admission and procoagulation factors, such as D-dimer (Pearson’s *r* = −0.317; *p* = 0.048).

The median time to clinical recovery in our cohort was 9.5 days (6–18). However, subjects with SZC at admission <50 µg/dL needed longer TR compared with those with zinc ≥ 50 µg/dL: 25 days (14–36) vs. 8 (5–14) days, respectively; *p* < 0.001. Moreover, SZC at admission was correlated negatively with the TR of the episode (Pearson’s *r* = −0.441; *p* < 0.001) (Figure 1D). Two alternative multivariable regression models were created, one using zinc as a quantitative continuous variable, and by adjusting for age, sex, severity, and Charlson comorbidity index, the TR was associated with low SZC at admission (beta-coefficient = −0.21 (95% CI= 0.316 to −0.097; *p* < 0.001)) (Appendix A). In an alternative model, adjusting serum zinc as a dichotomous variable (SZC < 50 µg/dL or ≥50 µg//dL) and adjusting by age, sex severity, and Charslon comorbidity index, serum zinc levels <50 µg/dL negatively impacted the TR (beta-coefficient = 14.1 (95% CI = 4.29–23.94; *p* = 0.004) (Appendix A).

Twenty-one individuals (9%) died during this period. SZC at admission was significantly higher among individuals who survived (62 µg/dL (52–72)) compared to those who died (49 µg/dL (42–53); *p* < 0.001). Individuals with SZC at admission <50 µg/dL had a mortality of 21%, which was significantly higher compared to the 5% mortality in individuals with zinc at admission ≥50 µg/dL (*p* < 0.001). We fitted a multivariable logistic regression model, including 249 patients with data for all variables (228 survivors and 21 non-survivors). When adjusting by age, sex, Charlson comorbidity index, and severity, the model showed an odds ratio (OR) for in-hospital death of 0.94 (95% CI = 0.899 to 0.982; *p* = 0.006) per unit increase of serum zinc at admission. In an alternative age-, sex-, severity-, and Charslon comorbidity index-adjusted model with the predictive variable SZC at admission <50 µg/dL, the adjusted OR for in-hospital death was 3.2 (95% CI = 1.01 to 10.12; *p* = 0.047). Other models, adjusting by individual mortality risk factors, along with age, sex, and severity, did not show differences with the final model for in-hospital mortality.

### 3.2. Impact of Cellular Zinc Content on SARS-CoV2 Expansion

Besides its impact on immune response, SZC might influence viral expansion [10]. Therefore, to obtain a wider picture of determinants involved in COVID-19 severity derived from SZC, we carried out assays in a cell culture to study the direct impact of cellular zinc content on SARS-CoV-2 multiplication. We used three different concentrations of zinc in the extracellular medium—0, 10, and 50 µM ZnSO_4_—to reproduce zinc deficiency, physiological zinc, and zinc supplementation, respectively. As expected, intracellular Zn^2+^ content changed in Vero E6 cells incubated for 30 min in different extracellular Zn^2+^ concentrations and monitored with Fluozin 3AM and Zinquin fluorescence probes. Confocal images showed that the FluoZin 3AM signal was mainly localized in the lysosomal compartment, while the Zinquin signal was intracellularly spread (Figure 2B). After 48 h incubation, different Zn^2+^ content solutions did not impact cell viability (Figure 2C). Importantly, extracellular zinc concentrations affected SARS-CoV-2 multiplication, as measured by qPCR from the cell culture supernatant at 48 h post-infection. SARS-CoV-2 multiplication was increased at 0 µM Zn^2+^ when compared to those values at 10 and 50 µM Zn^2+^ concentrations (Figure 2D). This indicates that zinc levels affect the SARS-CoV-2 life cycle in infected cells.

### 3.3. Assessment of Zinc Properties as a Potentiator of Chloroquine’s Antiviral Action

It has been suggested that Zn^2+^ may potentiate CQ antiviral activity, because CQ acts as a zinc ionophore [15]. Zinc supplementation, in combination with CQ, has been used to treat COVID-19 patients, showing benefits [16,17]. To test the potentiation effect against COVID-19, we carried out SARS-CoV-2 infections at different zinc concentrations in the absence or presence of 10 µM CQ; the concentration was chosen based on previously published effective concentrations [13,14]. Neither the CQ toxicity nor its antiviral activity were affected by zinc levels (Figure 3A,B). CQ caused a significant reduction in SARS-CoV-2 RNA genome copies compared to control cells (59.68% ± 18.46%; *p* < 0.05) (Figure 3B). Next, we addressed whether CQ increases the intracellular Zn^2+^ content by acting as a zinc ionophore, as was previously proposed [15]. For this, we evaluated the cytosolic Zn^2+^ levels in cells grown under different CQ concentrations, using flow cytometry analysis and FluoZin-3AM and Zinquin labels (Figure 3C). Notably, an increased zinc signal was observed in a CQ dose-dependent manner using FluoZin-3AM, but not with Zinquin (Figure 3C). This indicates that CQ modifies lysosomal Zn^2+^ content, but is not a zinc ionophore. As the described increase in lysosomal pH caused by CQ blocks autophagic flux [18] and inhibition of autophagy impairs SARS-CoV-2 replication [19], we studied the impact on autophagy of different zinc concentrations in the presence and the absence of CQ (Figure 3D–F). As expected, at 10 µM, CQ treatment autophagy blockade resulted in an increased LC3II/I ratio and p62 expression (Figure 3E,F). However, no significant effects on these values were observed at different Zn^2+^ levels in the absence or presence of CQ. Altogether, our results indicate that CQ and zinc do not potentiate each other.

## 4. Discussion

Our study demonstrates a correlation between serum zinc levels and COVID-19 outcome. SZC lower than <50 µg/dL at admission correlated with worse clinical presentation, longer time to recovery, and higher mortality. These results might suggest that SZC impacts COVID-19 severity, and its adjustment might also constitute an early therapeutic intervention point. These results are in accordance with the observational studies carried out by other groups with smaller cohorts [20,21]. Moreover, we show that SZC affects the SARS-CoV-2 life cycle in infected cells. This effect, contrary to what it has been previously suggested, does not seem to potentiate CQ activity.

The association between zinc and human health has been known for decades [1]. Due to poor nutrition and subsequent low zinc intake, ZD remains a major nutritional problem in multiple countries. In addition, elderly individuals are prone to ZD even in developed countries, where the incidence ranges from 15% to 31%, depending on the age and the country of study [22]. Older adults are the group at higher risk for severe symptoms and mortality from COVID-19 [23]. In our retrospective observational study, with 249 COVID-19 patients admitted to Hospital del Mar, at admission 23% of them had SZC lower than 50µg/dL, the cutoff associated with severe ZD and development of clinical signs [12].

At onset, higher levels of inflammatory markers, such as IL-6 and C-reactive protein, were present in low-SZC patients (Figure 1A,C). The prognostic value of IL-6 and C-reactive protein for COVID-19 severity has already been described [24]. In this work, we add SCZ as a novel early predictor for COVID-19 outcome. In addition, we observed a robust negative correlation between zinc levels and the highest peak of IL-6 highest peak (Figure 1B). These results suggest that COVID-19 patients with low SZC have an exacerbated immune response. ZD is known to be associated with proinflammatory responses at infection, showing higher reactive oxygen species production and inflammatory markers [25,26]. An imbalance in cytokine production by cells of both innate and adaptive immunity has also been reported [2,3]. Several clinical trials using zinc supplementation have been carried out to prevent and treat infections and inflammatory conditions [1,2]. Zinc supplementation decreases the incidence of infection in elderly and improves cytokine imbalance and oxidative stress markers [27,28]. Thus, the idea of supplementing low SZC COVID-19 patients with zinc in order to balance what has been called the cytokine storm caused by SARS-CoV-2 is attractive. Reassuringly, zinc supplementation for the common cold caused by rhino- and coronaviruses has been proven to reduce its duration and symptoms [8]. Serum zinc content is a common and recommended biomarker to assess zinc status and the risk of zinc deficiency morbidity [12]. Nevertheless, it is also known that upon acute infection, IL-6 promotes zinc internalization in hepatocytes [29]. Therefore, we cannot discard that the hypozincemia observed in COVID-19 patients might be caused and worsened by a negative IL-6 production loop. As a consequence, one limitation of our retrospective study is that our SZC measurements might not indicate the zinc nutritional status of COVID-19 patients before infection.

In addition to the impact of zinc in the modulation of the antiviral immune response, zinc has also been shown to have direct antiviral action [9]. We have analyzed in vitro the impact of zinc homeostasis in SARS-CoV-2 infection. Our results indicate that hypozincemia favors viral expansion in the infected cell (Figure 2C). These results would support that the poor clinical outcome observed in low-SZC patients is caused by the effect of low zinc availability on both, inducing immune imbalance and increasing viral load by promoting viral expansion in the infected cell. However, it is noteworthy that our in vitro studies did not show a replication blockade in zinc supplementation conditions (50 µM, Figure 2C), suggesting the need of a zinc ionophore to further increase cytosolic zinc levels in order to block viral replication, as previously shown in vitro for herpes simplex, picornavirus, arterivirus, and SARS-CoV (9,10).

During the beginning of the pandemic CQ was prescribed as a first-line treatment in the acute presentation. The rationale was that CQ has been claimed to be a novel zinc ionophore [15], and had shown antiviral effects against SARS-CoV-2 in vitro [13,14]. However, clinical trials failed to demonstrate its beneficial effects against COVID-19 [30,31]. Therefore, it was suggested that zinc supplementation could potentiate its antiviral activity [32]. In fact, there was an observational study treating COVID-19 patients with CQ supplemented with zinc that showed a reduction in mortality [16,17]. In our results in vitro, cellular zinc content did not modify CQ cytotoxicity, the autophagic flux blockade, or its antiviral action against SARS-CoV-2. Moreover, we monitored intracellular zinc content both with Fluozin-3AM, a probe previously done by Xue and colleagues that is retained mainly in lysosomes [15], and with Zinquin, which presents more general intracellular staining. We conclude that CQ is not a zinc ionophore, as previously claimed, because its effect on cellular zinc content is restricted to the lysosomal compartment, probably by altering zinc transport at this specific organelle (Figure 3). Thus, our study does not support the mechanistic rationale of supplementing CQ treatments with zinc. The positive results observed by Carlucci and collaborators in COVID-19 patients treated with zinc sulfate should not be attributed to a CQ potentiation effect [17].

This study has some limitations, since we cannot prove causality. Thus, zinc levels at admission could be impacted by other factors, such as acute phase reactants themselves [27]. Moreover, this is a single center study with a limited sample size. However, the study also has important strengths, since it has been conducted in a hospital with a large population area that is a representative area of an European city affected by the COVID-19 pandemic. All patients were attended in the COVID-19 unit under the same guidelines, with well-protocolized clinical procedures and with a centralized database that made the management of data uniform, reducing the weakness of the clinical data collection. This makes the quality and quantity of data coming from the COVID-19 unit reliable and homogeneous.

## 5. Conclusions

This work aims to focus clinical attention on serum zinc content in COVID-19 patients. Our analysis has shown a robust correlation between low SZC and COVID-19 severity and mortality. The cause is likely to be a combination of immune system imbalance and a direct benefit of viral replication. Thus, we propose SZC as a novel and additional parameter to predict COVID-19 outcome. It is then urgent to start clinical trials supplementing patients with low SZC at admission with zinc to reestablish zinc homeostasis. It should be also recommended to promote zinc supplementation programs targeted to people at risk of zinc deficiency, such as the elderly, in order to reduce COVID-19’s severity.

## Figures and Tables

**Figure 1 nutrients-13-00562-f001:**
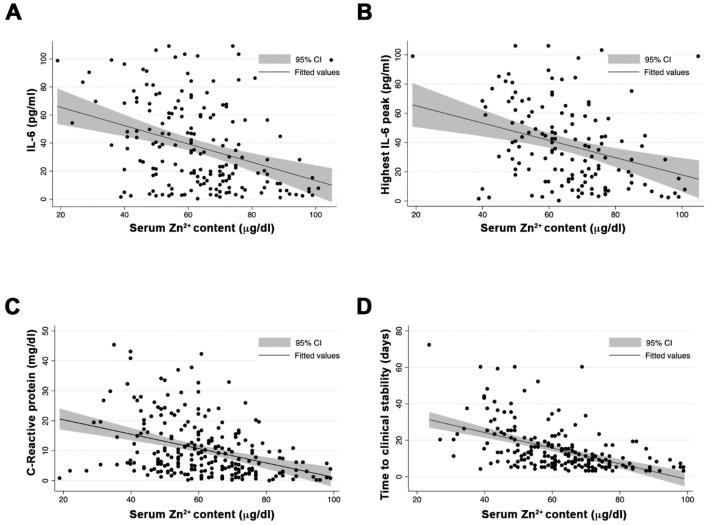
Correlation between serum zinc levels at admission, inflammatory markers, and clinical outcome. (**A**) Zinc and IL-6 at admission. (**B**) Zinc at admission and highest value of IL-6 during the episode. (**C**) Zinc and C-reactive protein at admission. (**D**) Serum zinc content at admission and time to clinical stability.

**Figure 2 nutrients-13-00562-f002:**
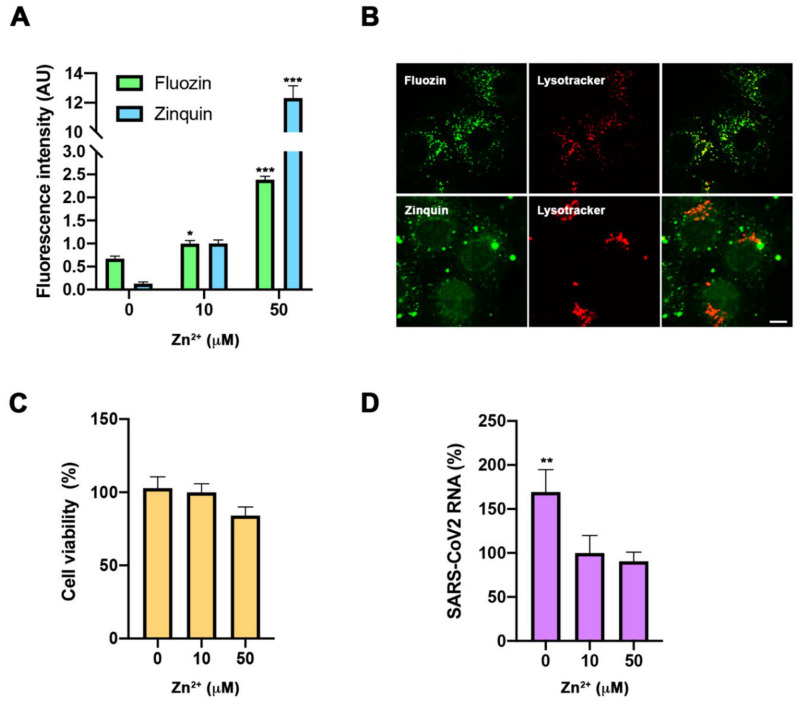
Evaluation of zinc homeostasis in SARS-CoV-2 infection. (**A**,**B**). Intracellular zinc content measurement using FluoZin-3AM and Zinquin probes in Vero E6 cells. (**A**). Flow cytometry in cells incubated for 30 min with 0, 10, and 50 µM extracellular Zn^2+^ content. Intensity expressed in arbitrary units (AU) (*n* = 3; * *p* < 0.05, *** *p* < 0.001; Bonferroni-corrected one-way ANOVA compared to 0 Zn^2+^). (**B**). Confocal images of living cells incubated with FluoZin-3AM (up) or Zinquin (bottom) and Lysotyracker in 50 µM Zn^2+^ extracellular medium. Scale bar = 10 µm. (**C**). Viability MTT assay in cells incubated with 0, 10, and 50 µM extracellular Zn^2+^ content for 48 h. Data expressed in percentage compared to control condition (10 µM Zn^2+^) (*n* = 12; Bonferroni-corrected one-way ANOVA compared to 10 µM Zn^2+^). (**D**). Quantification in supernatant of viral RNA copies by qPCR in cells infected with SARS-CoV-2 and collected at 48 h. Data expressed in percentage compared to control condition (10 µM Zn^2+^) (*n* = 3; ** *p* < 0.01; Bonferroni ANOVA compared to 10 µM Zn^2+^).

**Figure 3 nutrients-13-00562-f003:**
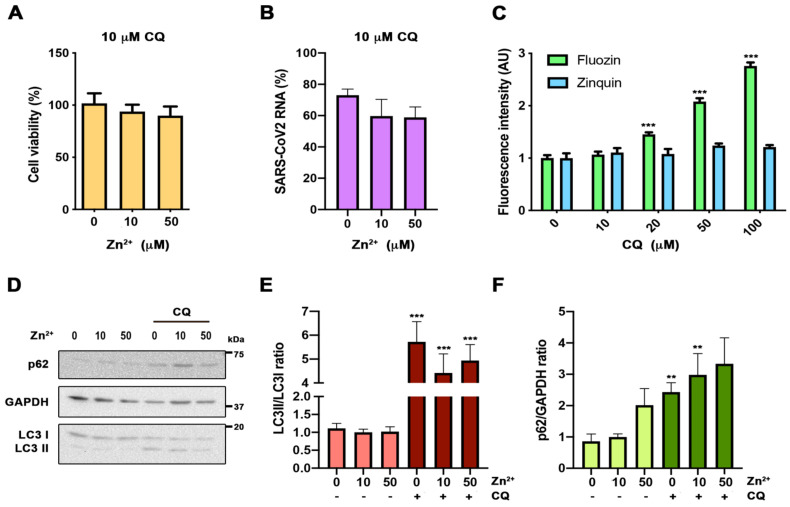
Evaluation of zinc potentiation of chloroquine antiviral action. (**A**) Viability MTT assay in cells incubated with 10 µM chloroquine (CQ) in 0, 10, and 50 µM Zn^2+^ content for 48 h (*n* = 15; Bonferroni-corrected one-way ANOVA compared to 10 µM Zn^2+^). (**B**) Quantification in the supernatant of viral RNA copies by qPCR in cells infected with SARS-CoV-2 treated with 10 µM CQ in 0, 10, and 50 µM Zn^2+^, and collected at 48 h. Data expressed in percentage compared to control condition (10 µM Zn^2+^ without CQ, Figure 2D) (*n* = 3; Bonferroni-corrected one-way ANOVA compared to 10 µM CQ in 10 µM Zn^2+^). (**C**) Flow cytometry in cells incubated for 30 min with 0, 10, and 50 µM extracellular Zn^2+^ content using FluoZin-3AM and Zinquin probes. Intensity expressed in arbitrary units (AU) (*n* ≥ 4; *** *p* < 0.001; Bonferroni-corrected one-way ANOVA compared to 0 µM Zn^2+^ from each probe). (**D**–**F**) Western blot analysis from cells incubated for 24 h in 0, 10, and 50 µM Zn^2+^ with or without 10 µM CQ. Antibodies against microtubule-associated proteins 1A/1B light chain 3B (LC3), p62, and glyceraldehyde 3-phosphate dehydrogenase (GAPDH). (**D**) Representative blot. (**E**,**F**) Quantification of the bands for LC3 ((**E**) *n* ≥ 10) and p62 ((**F**) *n* ≥ 7) (Bonferroni-corrected one-way ANOVA compared to 10 µM Zn^2+^ or 10 µM CQ in 10 µM Zn^2+^; ** *p* < 0.01, *** *p* < 0.001 unpaired *t*-test comparing conditions with and without CQ).

**Table 1 nutrients-13-00562-t001:** Main clinical characteristics of the cohort. Differences between individuals with plasma zinc at admission at <50 mcg/dL and ≥50 mcg/dL.

	Overall		<50 µg/dL		≥50 µg/dL		*p*-Value
Cohort Characteristics	*n* = 249			*n* = 58	*n* = 191		
Median age, years (IQR)	65	(54–75)	65	(59–75)	64	(53–74)	0.363
Male sex (%)	127	−51%	30	−51%	97	−51%	0.929
**Comorbidities**							
Current smoker (%)	23	−9.30%	4	−7%	19	−10%	0.482
Hypertension (%)	141	−56%	33	−57%	108	−57%	0.962
Diabetes mellitus (%)	53	−21%	17	−29%	36	−19%	0.088
Chronic lung disease (%)	22	−9%	9	−16%	13	−7%	0.041
Chronic heart disease (%)	37	−14%	11	−19%	26	−14%	0.315
Chronic renal disease (%)	70	−12%	22	−38%	48	−25%	0.058
Chronic liver disease (%)	18	−7%	1	−2%	17	−9%	0.065
Dementia (%)	8	−3%	2	−3%	6	−3%	0.908
HIV infection (%)	5	−2%	1	−2%	4	−2%	0.865
Active cancer (%)	7	−3%	3	−5%	4	−2%	0.214
ACE inhibitors (%)	61	−24%	12	−21%	49	−26%	0.441
Charlson Comorbidy Index, median (IQR)	1	(0–3)	1	(0–3)	1	(0–2)	0.247
**Symptoms at onset**							
Median days since syntoms onset (IQR)	7	(4–9)	6	(3–7)	7	(5–10)	0.005
Dyspnoea (%)	151	−60%	42	−72%	109	−57%	0.036
Fever (%)	203	−81%	45	−77%	158	−82%	0.377
Cough (%)	198	−79%	43	−74%	155	−81%	0.246
Diarrhea (%)	68	−27%	12	−21%	56	−29%	0.196
**Clinical markers at onset**							
Median C-reactive protein mg/dL (IQR)	7.5	(3.5–15.2)	14.6	(5–24)	7	(3–13)	0.037
Median lymphocyte count/mL (IQR)	1.02	(0.71–1.4)	0.82	(0.57–1.18)	1.1	(0.78–1.48)	0.598
Median interleukin (IL)-6 pg/mL (IQR)	42	(15–89)	77	(39–145)	32	(11–71)	<0.001
Median lactate dehydrogenase UI/l (IQR)	288	(241–345)	356	(275–483)	274	(231–362)	0.001
Median D-dimer UI/l (IQR)	800	(470–1450)	935	(540–1700)	800	(460–1215)	0.048
Median PaO_2_FiO_2_ ratio (IQR)	177	(100–299)	124	(91–181)	219	(106–314)	<0.001
Median modified early warning score (MEWS) (IQR)	2	(1–3)	2	(2–3)	2	(1–3)	0.005
Median serum zinc, mcg/mL (IQR)	61	(50–71)	43	(39–48)	66	(58–74)	<0.001
**Treatment**							
Hydroxycholoroquine (%)	248	−99.50%	58	−100%	190	−99%	0.372
Azythromicin (%)	231	−95%	57	−95%	174	−95%	0.766
Tocilizumab (%)	55	−23%	23	−40%	32	−17%	<0.001
Dexamethasone (%)	64	−26%	24	−41%	40	−21%	<0.001
Methylprednisolone (%)	59	−24%	26	−45%	33	−17%	<0.001
**Clinical Outcomes**							
Median Time to clinical recovery days (IQR)	10	(6–18)	25	(14–36)	8	(5–14)	<0.001
Intensive care unit (ICU) admission (%)	70	−28%	36	−62%	34	−18%	<0.001
Death (%)	21	−9%	12	−21%	9	−5%	<0.001

IQR (Interquartile Range).

## Data Availability

The data presented in this study are available on request from the corresponding author. The data are not publicly available due to Ethics Comitee restriction for personal data protection.

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
