# Peer review of "Low Zinc Levels at Admission Associates with Poor Clinical Outcomes in SARS-CoV-2 Infection"

_nutrients, 2021, doi:10.3390/nu13020562_

Round 1

Reviewer 1 Report

The paper of Guerri-Fernandez group deals with the serum zinc levels in patients with SARS-CoV-2. A group of 249 patients from Hospital del Mar in Barcelona with a positive qPCR test was taken into account. The paper is very interesting; results are well discussed; numerous valuable thesis were propounded. However, I have some doubts that I would like to see answered before the manuscript has conditions to be accepted:

  1. the quality of the figures is very poor. All of them should be improved in the final version.
  2. I am curious why concentration 50 µg/dL was set as differentiating value? Is it physiological serum zinc level? There should be an explanation in the manuscript.
  3. Almost all patients were treated with hydroxychloroquine. Many studies suggest it is a Zn(II) ionophore (I saw a conclusion in this paper that CQ is not a zinc ionophore, but in my opinion, further studies should be performed to confirm this conclusion). Nevertheless, does iontophoretic properties of hydroxychloroquine impact the zinc serum concentration?

Reviewer 2 Report

This observational study provides further insight into the possible role of trace elements in the progress of COVID-19 disease. In addition to analysing serum samples from affected patients, the authors also investigated the possible influence of different zinc concentrations on the expression patterns of SARS-CoV 2 viruses at the cellular level. These data provide valuable insights into the antioxidant mechanisms of the trace element zinc and imply more indications for the currently discussed intervention approach of supplementation of particularly deficient patients. Despite the valuable work, some modifications should still be made so that the content is presented to the reader in the best possible way:

Unfortunately, the quality of figure 1-3 wasn´t sufficient to evaluate the content. Therefore, I would ask the authors to increase the resolution of the figures so that the illustrations become comprehensible for the reader.

  1. Abstract

Line 30. The authors wrote “Importantly, zinc deficiency (ZD) is a common condition in elderly and individuals with chronic diseases, two groups with more severe COVID-19 outcomes”.

The authors should add "an increased risk" for a severe course of the disease, because not every older person automatically takes a worse clinical course.

  1. Materials and Methods

In line 81-83 the authors wrote: “COVID-19 was defined as a SARS-CoV-2 infection confirmed by quantitative PCR (qPCR) performed in nasopharyngeal samples obtained by trained personnel at hospital admission, and/or by fulfilling clinical diagnostic criteria.

When reading this paragraph, the question arises whether all patients included in the study had a positive PCR detection for SARS-CoV2 or whether patients without pathogen detection were also included?

Furthermore, the authors wrote: “Laboratory workouts were systematized with an at-admission protocol that included a fasting blood draw complete kidney and liver profile, electrolytes, blood count, coagulation profile, inflammatory markers (interleukin-6 (IL-6), serum ferritin), D-dimer, myo-cardial enzymes, zinc and selenium levels. Follow-up laboratory workout in the clinical setting included C-Reactive protein and IL-6, that made possible to determine IL-6 peak.”

            Because it is not clear from the other paragraphs, the following questions arise:

Which method was used to determine the serum zinc concentrations?

How many samples are available per patient? One sample each at the time of admission or were other samples taken during the inpatient stay?

Are there any results on the selenium quantifications? Several papers in the literature suggest an influence of selenium on clinical outcome after COVID-19.

The authors wrote: “In a multivariable linear regression model adjusted by age, sex, severity and Charlson comorbidity index TR was associated with low SZC at admission (beta-coeffi-cient -0.21 (95% CI-0.316 to-0.097; p<0.001)). In an alternative model adjusting by age, sex severity and Charslon comorbidity index serum zinc levels <50 μg/dl negatively impacted on TR (beta-coefficient 14.1 (95% CI 4.29 -23.94; p=0.004)

At this point, it is not entirely clear what constitutes the difference between the two models. Furthermore, there is no description of the different models and their informative value in the further script. For this purpose, a presentation might be useful that reveals which models were formed, which parameters were included, which (significant) differences were detected.

Furthermore, the index is called the " Charlson comorbidity index " and it might be helpful for the reader to have an explanation of the meaning, or which parameters are being considered.

  1. Results

To facilitate the understanding of the results, a subdivision into subcategories such as "Results of serum zinc analyses at the time of hospital admission", "Influence of cellular zinc content on SARS-CoV-2 multiplication" and "Assessment of zinc potentiation of the antiviral effect of chloroquine" might be helpful.

  1. Discussion

The authors wrote “Older adults are the group at higher risk for severe symptoms and mortality from COVID-19.”

By now, this fact is well known, but it is still useful to name a source for this statement.

Moreover, the authors wrote “In this work we add SCZ as a novel early predictor for COVID-19 outcome”.

The assumption that serum zinc concentrations could have a prognostic value/ impact on the clinical outcome in COVID-19 disease has also been made in other papers, for example by Jothimani et al. (COVID-19: Poor outcomes in patients with zinc deficiency; PMID: 32920234

) and Heller et al. (Prediction of survival odds in COVID-19 by zinc, age and selenoprotein P as composite biomarker; PMID: 33126054). This has not been addressed in the discussion. Are the results the same as those of the other studies? Are there any contradictions or differences?

  1. References

There are 27 references listed but only 26 are cited in the text.

Round 2

Reviewer 1 Report

All the issues have been carefully addressed.
I recommend the publication of this manuscript in its present form.

Reviewer 2 Report

With their work, the authors provide valuable further insights into the role of trace elements in the disease process of COVID-19 disease. Many thanks for the adjustments that have been made. The work has thus enhanced in quality. I wish you continued success in the medical care of patients and in research into this disease.